# CopulaGNN: Towards Integrating Representational and Correlational Roles of Graphs in Graph Neural Networks

**Jiaqi Ma**
School of Information
University of Michigan
jiaqima@umich.edu

**Bo Chang**
Department of Statistics
University of British Columbia
bchang@stat.ubc.ca

**Xuefei Zhang**
Department of Statistics
University of Michigan
xfzhang@umich.edu

**Qiaozhu Mei**
School of Information and Department of EECS
University of Michigan
qmei@umich.edu

## Abstract

Graph-structured data are ubiquitous. However, graphs encode diverse types of information and thus play different roles in data representation. In this paper, we distinguish the *representational* and the *correlational* roles played by the graphs in node-level prediction tasks, and we investigate how Graph Neural Network (GNN) models can effectively leverage both types of information. Conceptually, the representational information provides guidance for the model to construct better node features; while the correlational information indicates the correlation between node outcomes conditional on node features. Through a simulation study, we find that many popular GNN models are incapable of effectively utilizing the correlational information. By leveraging the idea of the copula, a principled way to describe the dependence among multivariate random variables, we offer a general solution. The proposed Copula Graph Neural Network (CopulaGNN) can take a wide range of GNN models as base models and utilize both representational and correlational information stored in the graphs. Experimental results on two types of regression tasks verify the effectiveness of the proposed method[1].

## 1 Introduction

Graphs, as flexible data representations that store rich relational information, have been commonly used in data science tasks. Machine learning methods on graphs (Chami et al., 2020), especially Graph Neural Networks (GNNs), have attracted increasing interest in the research community. They are widely applied to real-world problems such as recommender systems (Ying et al., 2018), social network analysis (Li et al., 2017), and transportation forecasting (Yu et al., 2017). Among the heterogeneous types of graph-structured data, it is worth noting that graphs usually play diverse roles in different contexts, different datasets, and different tasks. Some of the roles are relational, as a graph may indicate certain statistical relationships of connected observations; some are representational, as the topological structure of a graph may encode important features/patterns of the data; some are even causal, as a graph may reflect causal relationships specified by domain experts.

It is crucial to recognize the distinct roles of a graph in order to correctly utilize the signals in the graph-structured data. In this paper, we distinguish the *representational role* and the *correlational role* of graphs in the context of node-level (semi-)supervised learning, and we investigate how to design better GNNs that take advantage of both roles.

In a node-level prediction task, the observed graph in the data may relate to the outcomes of interest (e.g., node labels) in multiple ways. Conceptually, we call that the graph plays a representational

---

[1]The code is available at https://github.com/jiaqima/CopulaGNN.

role if one can leverage it to construct better feature representations. For example, in social network analysis, aggregating user features from one's friends is usually helpful (thanks to the well-known homophily phenomenon (McPherson et al., 2001)). In addition, the structural properties of a user's local network, e.g., structural diversity (Ugander et al., 2012) and structural holes (Burt, 2009; Lou & Tang, 2013), often provide useful information for making predictions about certain outcomes of that user. On the other hand, sometimes a graph directly encodes correlations between the outcomes of connected nodes, and we call it playing a *correlational role*. For example, hyper-linked Webpages are likely to be visited together even if they have dissimilar content. In spatiotemporal predictions, the outcome of nearby locations, conditional on all the features, may still be correlated. We note that the graph structure may provide useful predictive information through both roles but in distinct ways.

While both the representational and the correlational roles are common in graph-structured data, we find that, through a simulation study, many existing GNN models are incapable of utilizing the correlational information encoded in a graph. Specifically, we design a synthetic dataset for the node-level regression. The node-level outcomes are drawn from a multivariate normal distribution, with the mean and the covariance as functions of the graph to reflect the representational and correlation roles respectively. We find that when the graph only provides correlational information of the node outcomes, many popular GNN models underperform a multi-layer perceptron which does not consider the graph at all.

To mitigate this deficiency of GNNs, we propose a principled solution, the Copula Graph Neural Network (CopulaGNN), which can take a wide range of GNNs as the base model and improve their capabilities of modeling the correlational graph information. The key insight of the proposed method is that, by decomposing the joint distribution of node outcomes into the product of marginal densities and a copula density, the representational information and correlational information can be separately modeled. The former is modeled by the marginal densities through a base GNN while the latter is modeled by a Gaussian copula. The proposed method also enjoys the benefit of easy extension to various types of node outcome variables including continuous variables, discrete count variables, or even mixed-type variables. We instantiate CopulaGNN with normal and Poisson marginal distributions for continuous and count regression tasks respectively. We also implement two types of copula parameterizations combined with two types of base GNNs.

We evaluate the proposed method on both synthetic and real-world data with both continuous and count regression tasks. The experimental results show that CopulaGNNs significantly outperform their base GNN counterparts when the graph in the data exhibits both correlational and representational roles. We summarize our main contributions as follows:

1. We raise the question of distinguishing the two roles played by the graph and demonstrate that many existing GNNs are incapable of utilizing the graph information when it plays a pure correlational role.
2. We propose a principled solution, the CopulaGNN, to integrate the representational and correlational roles of the graph.
3. We empirically demonstrate the effectiveness of CopulaGNN compared to base GNNs on semi-supervised regression tasks.

## 2 RELATED WORK

There have been extensive existing works that model either the representational role or the correlational role of the graph in node-level (semi-)supervised learning tasks. However, there are fewer methods that try to model both sides simultaneously, especially with a GNN.

**Methods focusing on the representational role.** As we mentioned in Section 1, the graph can help construct better node feature representations by both providing extra topological information and guiding node feature aggregation. There have been vast existing studies on both directions, and among them we can only list a couple of examples. Various methods have been proposed to leverage the topological information of graph-structured data in machine learning tasks, such as graph kernels (Vishwanathan et al., 2010), node embeddings (Perozzi et al., 2014; Tang et al., 2015; Grover & Leskovec, 2016), and GNNs (Xu et al., 2018). Aggregating node features on an attributed graph has also been widely studied, e.g., through feature smoothing (Mei et al., 2008) or GNNs (Kipf

& Welling, 2016; Hamilton et al., 2017). In this work, we restrict our focus on the GNN models, which have been the state-of-the-art graph representation learning method on various tasks.

**Methods focusing on the correlational role.** On the other hand, there has also been extensive literature on modeling the dependence of variables on connected nodes in a graph. One group of methods is called the graph-based regularization (Zhu et al., 2003; Li et al., 2019), where it is assumed that the variables associated with linked objects change smoothly and pose an explicit similarity regularization among them. The correlational role of the graph is also closely related to undirected graphical models (Lauritzen, 1996; Jordan et al., 2004; Wainwright & Jordan, 2008). In graphical models, the edges in a graph provide a representation of the conditional (in)dependence structure among a set of random variables, which are represented by the node set of the graph. Finally, there has been a line of research that combines graphical models with copulas and leads to more flexible model families (Elidan, 2010; Dobra et al., 2011; Liu et al., 2012; Bauer et al., 2012). Our proposed method integrates the benefits of copulas and GNNs to capture both the representational and correlational roles.

**Methods improving GNNs by leveraging the correlational graph information.** A few methods explicitly leverage the correlational graph information to improve the GNN training, but most of them focus on a classification setting (Qu et al., 2019; Ma et al., 2019). A recent study (Jia & Benson, 2020) that we have been aware of only lately shares a similar motivation to ours, yet our methodology differs significantly. In particular, Jia & Benson (2020) apply a multivariate normal distribution to model the correlation of node outcomes, which can be viewed as a special case of our proposed CopulaGNN when a Gaussian copula with normal marginals is used. Our method not only generalizes to other marginals (we show the effectiveness of some of them), but also has a more flexible parameterization on the correlation matrix of the copula distribution. In addition, we differ with these previous works by explicitly distinguishing the two roles of the graph in the data.

## 3 SIMULATING THE TWO ROLES OF THE GRAPH

In this section, we investigate, through a simulation study, the representational and correlational roles of the graph in the context of node-level semi-supervised learning.

### 3.1 NODE-LEVEL SEMI-SUPERVISED LEARNING

We start by formally introducing the problem of node-level semi-supervised learning. A graph is a tuple: $\mathcal{G} = (\mathcal{V}, \mathcal{E})$, where $\mathcal{V} = \{1, 2, \ldots, n\}$ is the set of $n$ nodes; $\mathcal{E} \in \mathcal{V} \times \mathcal{V}$ is the set of edges and let $s = |\mathcal{E}|$ be the number of edges. The graph is also associated with $\mathbf{X} \in \mathbb{R}^{n \times d}$ and $\boldsymbol{y} \in \mathbb{R}^n$, which are the node features and outcome labels. In the semi-supervised learning setting, we only observe the labels of $0 < m < n$ nodes. Without loss of generality, we assume the labels of nodes $\{1, 2, \ldots, m\}$ are observed and those of $\{m+1, \ldots, n\}$ are missing. Therefore, the label vector $\boldsymbol{y}$ can be partitioned as $\boldsymbol{y} = (\boldsymbol{y}_{\text{obs}}^T, \boldsymbol{y}_{\text{miss}}^T)^T$. The goal of a node-level semi-supervised learning task is to infer $\boldsymbol{y}_{\text{miss}}$ based on $(\boldsymbol{y}_{\text{obs}}, \mathbf{X}, \mathcal{G})$.

### 3.2 SYNTHETIC DATA

To simulate the representational and correlational roles of the graph, we first design a synthetic dataset by specifying the joint distribution of $\boldsymbol{y}$ conditional on $\mathbf{X}$ and $\mathcal{G}$. In particular, we let the joint distribution of the node outcomes take the form of $\boldsymbol{y}|\mathbf{X}, \mathcal{G} \sim \mathcal{N}(\boldsymbol{\mu}(\mathbf{X}, \mathcal{G}), \boldsymbol{\Sigma}(\mathcal{G}))$, for some $\boldsymbol{\mu}, \boldsymbol{\Sigma}$ to be specified. In this way, the graph $\mathcal{G}$ plays a representational role through $\boldsymbol{\mu}(\mathbf{X}, \mathcal{G})$ and a correlational role through $\boldsymbol{\Sigma}(\mathcal{G})$.

Specifically, we generate synthetic node-level regression data on a graph with $n$ nodes and $s$ edges (see Appendix A.1 for the whole procedure). We first randomly generate a feature matrix $\mathbf{X} \in \mathbb{R}^{n \times d_0}$. Assume $\boldsymbol{A}$ is the adjacency matrix of the graph, $\boldsymbol{D}$ is the degree matrix, and $\boldsymbol{L} = \boldsymbol{D} - \boldsymbol{A}$ is the graph Laplacian. Let $\tilde{\boldsymbol{A}} = \boldsymbol{A} + \boldsymbol{I}$ and $\tilde{\boldsymbol{D}} = \boldsymbol{D} + \boldsymbol{I}$. Given parameters $\boldsymbol{w}_y \in \mathbb{R}^{d_0}$, we generate the node label vector $\boldsymbol{y} \sim \mathcal{N}(\boldsymbol{\mu}, \boldsymbol{\Sigma})$, where, for some $\gamma > 0, \tau > 0$, and $\sigma^2 > 0$,

(a) $\boldsymbol{\mu} = \tilde{\boldsymbol{D}}^{-1}\tilde{\boldsymbol{A}}\mathbf{X}\boldsymbol{w}_y, \boldsymbol{\Sigma} = \sigma^2 \boldsymbol{I}$;
(b) $\boldsymbol{\mu} = \mathbf{X}\boldsymbol{w}_y, \boldsymbol{\Sigma} = \tau(\boldsymbol{L} + \gamma\boldsymbol{I})^{-1}$;

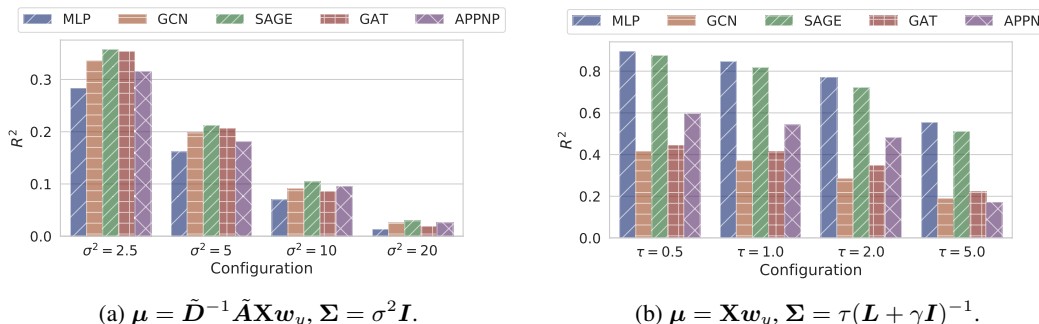

(a) $\boldsymbol{\mu} = \tilde{\boldsymbol{D}}^{-1}\tilde{\boldsymbol{A}}\mathbf{X}\boldsymbol{w}_y, \boldsymbol{\Sigma} = \sigma^2\boldsymbol{I}$.

(b) $\boldsymbol{\mu} = \mathbf{X}\boldsymbol{w}_y, \boldsymbol{\Sigma} = \tau(\boldsymbol{L} + \gamma\boldsymbol{I})^{-1}$.

Figure 1: The coefficient of determination $R^2$ (the higher the better) of GNNs and MLP when the graph plays (a) the representational role or (b) the correlational role. For each configuration, the results are aggregated from 100 trials. In (a), all GNNs outperform MLP; in (b), all GNNs underperform MLP.

(c) $\boldsymbol{\mu} = \tilde{\boldsymbol{D}}^{-1}\tilde{\boldsymbol{A}}\mathbf{X}\boldsymbol{w}_y, \boldsymbol{\Sigma} = \tau(\boldsymbol{L} + \gamma\boldsymbol{I})^{-1}$.

Depending on how $(\boldsymbol{\mu}, \boldsymbol{\Sigma})$ are configured, we get three types of synthetic data settings: (a), (b), and (c). Intuitively, the graph plays a pure representational role in setting (a) since the label of a node depends on the aggregated features of its local neighborhood and the node labels are independent conditional on the node features. In setting (b), the graph plays a pure correlational role; while the means of node labels only depend on their own node features, the node labels are still correlated conditional on the features, and the correlation is determined by the graph structure. Finally, setting (c) is a combination of (a) and (b) where the graph plays both representational and correlational roles.

In the rest of this section, we test the performance of a few widely used GNNs under setting (a) and (b) to examine their capabilities of utilizing the representational and correlational information. We defer the experimental results under setting (c) to Section 5.2 for ease of reading.

### 3.3 SIMULATION STUDY

**Simulation Setup.** We set the number of nodes $n = 300$, the number of edges $s = 5000$, and the feature dimension $d_0 = 10$. Elements of both $\boldsymbol{W}_g$ and $\boldsymbol{w}_y$ are generated from i.i.d. standard normal distribution. For setting (a), we vary $\sigma^2 \in \{2.5, 5, 10, 20\}$. For settings (b) and (c), we set $\gamma = 0.1$ and vary $\tau \in \{0.5, 1, 2, 5\}$. We test 4 common GNN models, GCN (Kipf & Welling, 2016), GraphSAGE (Hamilton et al., 2017) (denoted as SAGE), GAT (Veličković et al., 2018), and APPNP (Klicpera et al., 2018), as well as the multi-layer perceptron (MLP).

**Simulation Results.** First, we observe that all 4 types of GNNs outperform MLP under setting (a) (Figure 1a), where the graph plays a pure representational role. This is not surprising as the architectures of the GNNs encode a similar feature aggregation structure as the data. However, under setting (b) (Figure 1b) where the graph plays a pure correlational role, all 4 types of GNNs underperform MLP. This suggests that a majority of popular GNN models might be incapable of fully utilizing the correlational graph information.

Motivated by our findings in the simulation study, in the following section, we seek for methods that augment existing GNN models in order to better utilize both representational and correlational information in the graph.

## 4 COPULA GRAPH NEURAL NETWORK

In this section, we propose a principled solution called the Copula Graph Neural Network (CopulaGNN). At the core of our method is the application of copulas, which are widely used for modeling multivariate dependence. In the rest of this section, we first provide a brief introduction to copulas (more detailed expositions can be found in the monographs by Joe (2014) and Czado (2019)), then present the proposed CopulaGNN and its parameterization, learning, and inference.

## 4.1 INTRODUCTION TO COPULAS

In order to decompose the joint distribution of the labels $\boldsymbol{y}$ into representational and correlational components, we make use of *copulas*, which are widely used in multivariate statistics to model the joint distribution of a random vector.

**General formulation.** Sklar's theorem (Sklar, 1959) states that any multivariate joint distribution $F$ of a random vector $\boldsymbol{Y} = (Y_1, \ldots, Y_n)$ can be written in terms of one-dimensional marginal distributions $F_i(y) = \mathbb{P}(Y_i \leq y)$ and a copula $C : [0, 1]^n \to [0, 1]$ that describes the dependence structures among variables:

$$F(y_1, \ldots, y_n) = C(F_1(y_1), \ldots, F_p(y_n)).$$

In other words, one can decompose a joint distribution into two components: the marginals and the copula. Such decomposition allows a two-step approach to modeling a joint distribution: (1) learning the marginals $F_i$; (2) learning the copula $C$, where various parametric copula families are available. Furthermore, a copula $C$ can also be regarded as the Cumulative Distribution Function (CDF) of a corresponding distribution on the unit hypercube $[0, 1]^n$. Its copula density is denoted by $c(u_1, \ldots, u_n) := \partial^n C(u_1, u_2, \ldots, u_n)/\partial u_1 \cdots \partial u_n$. The Probability Density Function (PDF) of a random vector can be represented by its corresponding copula density. If the random vector $\boldsymbol{Y}$ is continuous, its PDF can be written as

$$f(\boldsymbol{y}) = c(u_1, \ldots, u_n) \prod_{i=1}^{n} f_i(y_i), \tag{1}$$

where $f_i$ is the PDF of $Y_i$, $u_i = F_i(y_i)$, and $c$ is the copula density. For discrete random vectors, the form of the Probability Mass Function (PMF) is more complex. See Appendix B.2 for details.

**Gaussian copula.** One of the most popular copula family is the *Gaussian copula*. When the joint distribution $F$ is multivariate normal with a mean of $\boldsymbol{0}$ and a covariance matrix of $\boldsymbol{\Sigma}$, the corresponding copula is the Gaussian copula:

$$C(u_1, u_2, \cdots, u_n; \boldsymbol{\Sigma}) = \Phi_n(\Phi^{-1}(u_1), \cdots, \Phi^{-1}(u_n); \boldsymbol{0}, \boldsymbol{R}),$$

where $\Phi_n(\cdot; \boldsymbol{0}, \boldsymbol{R})$ is the multivariate normal CDF, $\boldsymbol{R}$ is the correlation matrix of $\boldsymbol{\Sigma}$, and $\Phi^{-1}(\cdot)$ is the quantile function of the univariate standard normal distribution. Its copula density is

$$c(u_1, u_2, \ldots, u_n; \boldsymbol{\Sigma}) = (\det \boldsymbol{R})^{-1/2} \exp\left(-\frac{1}{2}\Phi^{-1}(\boldsymbol{u})^T(\boldsymbol{R}^{-1} - \boldsymbol{I}_n)\Phi^{-1}(\boldsymbol{u})\right),$$

where $\boldsymbol{I}_n$ is the identity matrix of size $n$ and $\boldsymbol{u} = (u_1, u_2, \ldots, u_n)$.

## 4.2 THE PROPOSED MODEL

Recall that our goal is to model both representational and correlational graph information in the conditional joint distribution of the node outcomes,

$$f(\boldsymbol{y}; \mathbf{X}, \mathcal{G}) = c(u_1, \ldots, u_n; \mathbf{X}, \mathcal{G}) \prod_{i=1}^{n} f_i(y_i; \mathbf{X}, \mathcal{G}), \tag{2}$$

which can be decomposed into the copula density and marginal densities. In this formulation, the representational information and correlational information are naturally separated into the marginal densities $f_i$, for $i = 1, \ldots, n$, and the copula density $c$, respectively. Note that both the marginal densities and the copula density are conditional on the node features $\mathbf{X}$ and the graph $\mathcal{G}$. Next, we need to choose a proper distribution family for each of these densities and parameterize the distribution parameters as functions of $\mathbf{X}$ and $\mathcal{G}$.

**Choice of distribution family and parameterization for the copula density.** For the distribution family, we choose the Gaussian copula as the copula family, $c(u_1, \ldots, u_n; \boldsymbol{\Sigma}(\mathbf{X}, \mathcal{G}; \boldsymbol{\theta}))$, where the form of $\boldsymbol{\Sigma}(\cdot; \boldsymbol{\theta})$ and the learnable parameters $\boldsymbol{\theta}$ remain to be specified. To leverage the correlational graph information, we draw a connection between the graph structure $\mathcal{G}$ and the covariance matrix $\boldsymbol{\Sigma}$ in the Gaussian copula density. Let $\boldsymbol{K} = \boldsymbol{\Sigma}^{-1}$ be the precision matrix; if two nodes $i$ and $j$ are not linked in the graph, we constrain the corresponding $(i, j)$-th entry in $\boldsymbol{K}$ to be 0. In other

words, the absence of an edge between nodes $i$ and $j$ leads to their outcome variables $y_i$ and $y_j$ being conditionally independent given all other variables. The motivation of parameterizing the precision matrix $\boldsymbol{K}$ instead of the covariance matrix $\boldsymbol{\Sigma}$ is closely related to undirected graphical models (Lauritzen, 1996; Jordan et al., 2004; Wainwright & Jordan, 2008), where the conditional dependence structure among a set of variables is fully represented by edges in an underlying graph. In our use case, we could view our assumption on $\boldsymbol{K}$ as a graphical model among random variables $(y_1, \ldots, y_n)$, where the underlying graph structure is known.

The conditional independence assumption has significantly reduced the number of non-zero entries in $\boldsymbol{K}$ to be estimated. However, without any further constraints, there are still $|\mathcal{E}|$ free parameters growing with the graph size, which can hardly be estimated accurately given only one observation of $(y_1, \ldots, y_m)$. In practice, we consider two ways of parametrizing $\boldsymbol{K}$ with fewer parameters.

*Two-parameter parametrization.* A rather strong but simple constraint is to assume the non-zero off-diagonal entries of $\boldsymbol{K}$ have the same value or they are proportional to the corresponding entries in the normalized adjacency matrix, and introduce two global parameters controlling the overall strength of correlation. For example, we could have $\boldsymbol{K} = \tau^{-1}(\boldsymbol{L} + \gamma\boldsymbol{I})$ as what we did in the simulation study in Section 3, or $\boldsymbol{K} = \beta(\boldsymbol{I}_n - \alpha\boldsymbol{D}^{-1/2}\boldsymbol{A}\boldsymbol{D}^{-1/2})$ as used in Jia & Benson (2020), where $(\tau, \gamma)$ or $(\alpha, \beta)$ are learnable parameters.

*Regression-based parametrization.* We further propose a more flexible parameterization that allows the non-zero entries in $\boldsymbol{K}$ to be estimated by a regressor taking node features as inputs. In pariticular, for any $(i, j)$-pair corresponding to a non-zero entry of $\boldsymbol{K}$, we set $\widehat{\boldsymbol{A}}_{i,j} = \text{softplus}(h(\boldsymbol{x}_i, \boldsymbol{x}_j; \boldsymbol{\theta}))$, where $h$ is a two-layer MLP that takes the concatenation of $\boldsymbol{x}_i$ and $\boldsymbol{x}_j$ as input and outputs a scalar. Let $\widehat{\boldsymbol{D}}$ be the degree matrix if we treat $\widehat{\boldsymbol{A}}$ as a weighted adjacency matrix, and we set the precision matrix $\boldsymbol{K} = \boldsymbol{I}_n + \widehat{\boldsymbol{D}} - \widehat{\boldsymbol{A}}$. This parameterization improves the flexibility on estimating $\boldsymbol{K}$ while keeping the number of learnable parameters $\boldsymbol{\theta}$ independent of the graph size $n$. It also ensures that $\boldsymbol{K}$ is positive-definite and thus invertible.

**Choice of distribution families and parameterization for the marginal densities.** One benefit of the copula framework is the flexibility on the choice of distribution families for the marginal densities. In this work, we choose the marginal densities to be normal distributions if the labels $\boldsymbol{y}$ are continuous variables, and we choose them to be Poisson distributions if the $\boldsymbol{y}$ are discrete. We denote the $i$-th marginal density function by $f_i(y_i; \boldsymbol{\eta}_i(\mathbf{X}, \mathcal{G}; \boldsymbol{\theta}))$ and the corresponding CDF by $F_i(y_i; \boldsymbol{\eta}_i(\mathbf{X}, \mathcal{G}; \boldsymbol{\theta}))$, where $\boldsymbol{\eta}_i(\cdot; \boldsymbol{\theta})$ denotes the distribution parameters to be specified.

If the $i$-th marginal distribution takes the form of a normal distribution $\mathcal{N}(\mu_i, \sigma_i^2)$, then $\boldsymbol{\eta}_i(\mathbf{X}, \mathcal{G}; \boldsymbol{\theta}) = (\mu_i(\mathbf{X}, \mathcal{G}; \boldsymbol{\theta}), \sigma_i^2(\mathbf{X}, \mathcal{G}; \boldsymbol{\theta}))$. We define $\mu_i(\mathbf{X}, \mathcal{G}; \boldsymbol{\theta})$ as the output of a base GNN model for node $i$, and $\sigma_i^2(\mathbf{X}, \mathcal{G}; \boldsymbol{\theta})$ as the $i$-th diagnoal element of the covariance matrix $\boldsymbol{\Sigma}(\mathbf{X}, \mathcal{G}; \boldsymbol{\theta})$ as we specified in the Gaussian copula. If the $i$-th marginal distribution takes the form of a Poisson distribution $\text{Pois}(\lambda_i)$, then $\boldsymbol{\eta}_i(\mathbf{X}, \mathcal{G}; \boldsymbol{\theta}) = \lambda_i(\mathbf{X}, \mathcal{G}; \boldsymbol{\theta})$, and we define $\lambda_i(\mathbf{X}, \mathcal{G}; \boldsymbol{\theta})$ as the output of a base GNN model for node $i$. Either way, the representational role of the graph is reflected in the location parameters ($\mu_i$ or $\lambda_i$) computed by a base GNN model. In practice, we can also choose other distribution families such as the log-normal or negative binomial, depending on our belief on the true distributions of the node outcomes. One can even choose different distribution families for different nodes simultaneously if necessary.

### 4.3 MODEL LEARNING AND INFERENCE

For simplicity of notation, we write $\boldsymbol{\eta}_i(\mathbf{X}, \mathcal{G}; \boldsymbol{\theta})$ and $\boldsymbol{\Sigma}(\mathbf{X}, \mathcal{G}; \boldsymbol{\theta})$ as $\boldsymbol{\eta}_i$ and $\boldsymbol{\Sigma}$ throughout this section.

**Model learning.** The model parameters $\boldsymbol{\theta}$ are learned by maximizing the log-likelihood on the observed node labels. Given the partition of $\boldsymbol{y}$, we can further partition the covariance matrix $\boldsymbol{\Sigma}$ accordingly:

$$\boldsymbol{y} = \begin{pmatrix} \boldsymbol{y}_{\text{obs}} \\ \boldsymbol{y}_{\text{miss}} \end{pmatrix} \text{ and } \boldsymbol{\Sigma} = \begin{pmatrix} \boldsymbol{\Sigma}_{00} & \boldsymbol{\Sigma}_{01} \\ \boldsymbol{\Sigma}_{10} & \boldsymbol{\Sigma}_{11} \end{pmatrix},$$

where $\boldsymbol{y}_{\text{obs}} = (y_1, \ldots, y_m)$ and $\boldsymbol{y}_{\text{miss}} = (y_{m+1}, \ldots, y_n)$. In other words, $\boldsymbol{\Sigma}_{00}$ and $\boldsymbol{\Sigma}_{11}$ are the covariance matrices of the observed and missing nodes. We further denote $u_i = F_i(y_i; \boldsymbol{\eta}_i)$ for $i = 1, \ldots, n$, $\boldsymbol{u}_{\text{obs}} = (u_1, \ldots, u_m)$, and $\boldsymbol{u}_{\text{miss}} = (u_{m+1}, \ldots, u_n)$; that is, $u_i$ is the probability integral transform of the $i$-th label $y_i$. According to Equation (1), the joint density can be written

as the product of the copula density and the marginal densities. Therefore the loss function, i.e., the negative log-likelihood, is

$$\mathcal{L}(\boldsymbol{\theta}) = -\log f(\boldsymbol{y}_{\text{obs}}; \mathbf{X}, \mathcal{G}) = -\log c(\boldsymbol{u}_{\text{obs}}; \boldsymbol{\Sigma}_{00}) - \sum_{i=1}^{m} \log f_i(y_i; \boldsymbol{\eta}_i). \tag{3}$$

The parameters $\boldsymbol{\theta}$ are learned end-to-end using standard optimization algorithms such as Adam (Kingma & Ba, 2015).

**Model inference.** At inference time, we are interested in the conditional distribution $f(\boldsymbol{y}_{\text{miss}}|\boldsymbol{y}_{\text{obs}}; \mathbf{X}, \mathcal{G})$. The inference of the conditional distribution can be done via sampling. Since $f(\boldsymbol{y}; \mathbf{X}, \mathcal{G})$ is modeled by the Gaussian copula, we have

$$\begin{pmatrix} \Phi^{-1}(\boldsymbol{u}_{\text{obs}}) \\ \Phi^{-1}(\boldsymbol{u}_{\text{miss}}) \end{pmatrix} \sim \mathcal{N}\left( \begin{pmatrix} \mathbf{0} \\ \mathbf{0} \end{pmatrix}, \begin{pmatrix} \boldsymbol{R}_{00} & \boldsymbol{R}_{01} \\ \boldsymbol{R}_{10} & \boldsymbol{R}_{11} \end{pmatrix} \right),$$

where $\boldsymbol{R}$ is the correlation matrix corresponding to the covariance matrix $\boldsymbol{\Sigma}$. By the property of the multivatiate normal distribution, the conditional distribution of $\Phi^{-1}(\boldsymbol{u}_{\text{miss}})|\Phi^{-1}(\boldsymbol{u}_{\text{obs}})$ is also multivatiate normal:

$$\Phi^{-1}(\boldsymbol{u}_{\text{miss}})|\Phi^{-1}(\boldsymbol{u}_{\text{obs}}) \sim \mathcal{N}\left( \boldsymbol{R}_{10}\boldsymbol{R}_{00}^{-1}\Phi^{-1}(\boldsymbol{u}_{\text{obs}}), \boldsymbol{R}_{11} - \boldsymbol{R}_{10}\boldsymbol{R}_{00}^{-1}\boldsymbol{R}_{01} \right). \tag{4}$$

This provides a way to draw samples from $f(\boldsymbol{y}_{\text{miss}}|\boldsymbol{y}_{\text{obs}}; \mathbf{X}, \mathcal{G})$, which we describe in Algorithm 1.

---

**Algorithm 1:** Model inference by sampling.

---

**Input:** The node features $\mathbf{X}$, the graph $\mathcal{G}$, the observed node labels $y_1, \ldots, y_m$, the learned parameters $\boldsymbol{\theta}$, the marginal CDF functions $F_i(\cdot; \boldsymbol{\eta}_i(\mathbf{X}, \mathcal{G}; \boldsymbol{\theta}))$ their inverse $F_i^{-1}(\cdot; \boldsymbol{\eta}_i(\mathbf{X}, \mathcal{G}; \boldsymbol{\theta}))$ for $i = 1, \ldots, n$, and the number of samples $L$.
**Output:** Predicted missing node labels $\hat{y}_i, i = m + 1, \ldots, n$.

1   **for** $i = m + 1, m + 2, \ldots, n$ **do**
2     $\hat{y}_i \leftarrow 0$;
3   **for** $i = 1, 2, \ldots, m$ **do**
4     $u_i \leftarrow F_i(y_i; \boldsymbol{\eta}_i(\mathbf{X}, \mathcal{G}; \boldsymbol{\theta}))$;
5     $z_i \leftarrow \Phi^{-1}(u_i)$;
6   $\boldsymbol{z}_{\text{obs}} \leftarrow [z_1, \ldots, z_m]^T$;
7   $\boldsymbol{R} \leftarrow$ the correlation matrix corresponding to $\boldsymbol{\Sigma}(\mathbf{X}, \mathcal{G}; \boldsymbol{\theta})$;
8   $\boldsymbol{\mu}_{\text{cond}} \leftarrow \boldsymbol{R}_{10}\boldsymbol{R}_{00}^{-1}\boldsymbol{z}_{\text{obs}}$;
9   $\boldsymbol{\Sigma}_{\text{cond}} \leftarrow \boldsymbol{R}_{11} - \boldsymbol{R}_{10}\boldsymbol{R}_{00}^{-1}\boldsymbol{R}_{01}$;
10 **for** $\ell = 1, 2, \ldots, L$ **do**
11     $[z_{m+1}^\ell, \ldots, z_n^\ell]^T \sim \mathcal{N}(\boldsymbol{\mu}_{\text{cond}}, \boldsymbol{\Sigma}_{\text{cond}})$;
12     **for** $i = m + 1, m + 2, \ldots, n$ **do**
13       $y_i^\ell \leftarrow F_i^{-1}(\Phi(z_i^\ell); \boldsymbol{\eta}_i(\mathbf{X}, \mathcal{G}; \boldsymbol{\theta}))$;
14       $\hat{y}_i \leftarrow \hat{y}_i + y_i^\ell/L$;

15 **return** $\hat{y}_i, i = m + 1, \ldots, n$;

---

**Scalability.** Finally, we make a brief remark on the scalability of the proposed method. During model learning, the calculation of $\log c(\boldsymbol{u}_{\text{obs}}; \boldsymbol{\Sigma}_{00})$ in Eq. (3) involves the log-determinant of the precision matrix $\boldsymbol{K}$ and its submatrix. During model inference, both the conditional mean and variance involve the evaluation of $\boldsymbol{R}_{00}^{-1}$, but they can be transformed (via Schur complement) in a form that only requires solving a linear system with a submatrix of $\boldsymbol{K}$ as the coefficients. Thanks to the sparse structure of $\boldsymbol{K}$, both (one-step) learning and inference can be made efficient with a computation cost linear in the graph size if $\boldsymbol{K}$ is well-conditioned. Jia & Benson (2020) provide an introduction of numerical techniques that can efficiently calculate the log-determinant (and its gradients) of a sparse matrix, as well as the solution of a linear system with sparse coefficients, which can be directly applied to accelerate our proposed method. In addition, as real-world graphs often present community structures, we can further improve the scalability of the proposed method by enforcing a block structure for the precision matrix.

Table 1: Experiment results on the synthetic data under setting (c) as described in Sections 3.2 and 3.3. The average test $R^2$ from 100 trials is reported (the larger the better). The asterisk markers, *, **, and ***, indicate the difference between a variant of CopulaGNN and its GNN base model is statistically significant by a pairwise $t$-test at significance levels of 0.1, 0.05, and 0.01, respectively. The ($\pm$) error bar denotes the standard error of the mean.

|  | $\tau = 0.5$ | $\tau = 1.0$ | $\tau = 2.0$ | $\tau = 5.0$ |
|---|---|---|---|---|
| MLP | $0.624 \pm 0.011$ | $0.549 \pm 0.014$ | $0.437 \pm 0.018$ | $0.193 \pm 0.020$ |
| GCN | $0.673 \pm 0.022$ | $0.563 \pm 0.034$ | $0.384 \pm 0.055$ | $0.174 \pm 0.032$ |
| $\alpha\beta$-C-GCN | $0.669 \pm 0.024$ | $0.568 \pm 0.033$ | $0.408 \pm 0.050^*$ | $0.200 \pm 0.035$ |
| R-C-GCN | $0.706 \pm 0.017^{***}$ | $0.617 \pm 0.023^{***}$ | $0.489 \pm 0.034^{***}$ | $0.217 \pm 0.029^*$ |
| SAGE | $0.733 \pm 0.013$ | $0.644 \pm 0.020$ | $0.507 \pm 0.030$ | $0.262 \pm 0.025$ |
| $\alpha\beta$-C-SAGE | $0.741 \pm 0.013^{**}$ | $0.650 \pm 0.019$ | $0.518 \pm 0.029^*$ | $0.281 \pm 0.024^{**}$ |
| R-C-SAGE | $0.754 \pm 0.010^{***}$ | $0.665 \pm 0.017^{**}$ | $0.540 \pm 0.024^{**}$ | $0.290 \pm 0.022^*$ |

## 5 EXPERIMENTS

### 5.1 GENERAL SETUP

We instantiate CopulaGNN with either GCN or GraphSAGE as the base GNN models, and implement both the two-parameter parameterization (the $(\alpha, \beta)$ parameterization, denoted by "$\alpha\beta$-C-", where "C" stands for copula) and the regression-based parameterization (denoted by "R-C-"). In combination, we have four variants of CopulaGNN: $\alpha\beta$-**C-GCN**, **R-C-GCN**, $\alpha\beta$-**C-SAGE**, and **R-C-SAGE**. When the outcome is a continuous variable, the normal margin is used; and when the outcome is a count variable, the Poisson margin is used. In particular, in the former case, the $\alpha\beta$-C-GNN degenerates to the Correlation GNN proposed by Jia & Benson (2020). We compare different variants of CopulaGNN with their base GNN counterparts, as well as an MLP model, on two types of regression tasks: continuous outcome variables and count outcome variables. More experiment details can be found in Appendix A.3.

### 5.2 REGRESSION WITH CONTINUOUS OUTCOME VARIABLES

We use two groups of datasets with continuous outcome variables. The first group is the synthetic data of setting (c) as described in Sections 3.2 and 3.3, where a graph provides both representational and correlational information. The second group includes four regression tasks constructed from the U.S. Election data (Jia & Benson, 2020). We use the coefficient of determination $R^2$ to measure the model performance.

**Results.** For the synthetic datasets (Table 1), we vary the value of $\tau$, which controls the overall magnitude of the label covariance. Unsurprisingly, as $\tau$ increases, the labels become noisier and the test $R^2$ of all models decreases. In all configurations, R-C-GCN and R-C-SAGE respectively outperform their base model counterparts, GCN and SAGE, by significant margins. This verifies the effectiveness of the proposed method when the graph provides both representational and correlational information. Another interesting observation is that GCN outperforms MLP when $\tau$ is small (0.5 and 1.0), but underperforms MLP when $\tau$ becomes large (2.0 and 5.0), whereas R-C-GCN consistently outperforms MLP. Note that $\tau$ can also be viewed as the tradeoff between the representational role and the correlational role served by the graph. The correlational role of the graph will have more influence on the outcome variables when $\tau$ becomes larger. This explains the intriguing observation: GCN fails to utilize the correlational information and its advantages on the representational information diminish as $\tau$ increases.

For the U.S. Election dataset (Table 2), we observe that all variants of CopulaGNN significantly outperform their base GNN counterparts. It is interesting that the simpler two-parameter parameterization outperforms the regression-based parameterization in most setups on this dataset. One possible explanation is that the outcome variables that are connected in the graph tend to have strong correlations, since adjacent counties usually have similar statistics. This is indeed suggested by Jia & Benson (2020). The Unemployment task in particular, where the two-parameter parameterization appears to have the largest advantage, is shown to have the strongest correlation.

Table 2: Experiment results on regression tasks of the U.S. Election dataset (with continuous outcome variables). The average test $R^2$ from 10 trials is reported (the larger the better). The asterisk markers and the ($\pm$) error bar indicate the same meaning as in Table 1.

|  | Education | Election | Income | Unemployment |
|---|---|---|---|---|
| MLP | $0.660 \pm 0.004$ | $0.400 \pm 0.003$ | $0.597 \pm 0.006$ | $0.400 \pm 0.004$ |
| GCN | $0.418 \pm 0.004$ | $0.472 \pm 0.002$ | $0.607 \pm 0.003$ | $0.572 \pm 0.010$ |
| $\alpha\beta$-C-GCN | $0.452 \pm 0.001$*** | $0.558 \pm 0.002$*** | $0.635 \pm 0.001$*** | $0.750 \pm 0.004$*** |
| R-C-GCN | $0.454 \pm 0.002$*** | $0.578 \pm 0.005$*** | $0.661 \pm 0.001$*** | $0.654 \pm 0.009$*** |
| SAGE | $0.677 \pm 0.004$ | $0.565 \pm 0.005$ | $0.714 \pm 0.006$ | $0.628 \pm 0.005$ |
| $\alpha\beta$-C-SAGE | $0.709 \pm 0.003$*** | $0.700 \pm 0.003$*** | $0.775 \pm 0.003$*** | $0.813 \pm 0.004$*** |
| R-C-SAGE | $0.707 \pm 0.003$*** | $0.692 \pm 0.005$*** | $0.763 \pm 0.003$*** | $0.695 \pm 0.003$*** |

Table 3: Experiment results on regression tasks with count outcome variables. The average test $R^2$ (deviance) from 50 trials is reported (the larger the better). The asterisk markers and the ($\pm$) error bar indicate the same meaning as in Table 1.

|  | EMNLP | Wiki-Chameleon | Wiki-Squirrel |
|---|---|---|---|
| MLP | $0.125 \pm 0.006$ | $0.347 \pm 0.008$ | $0.439 \pm 0.004$ |
| GCN | $0.609 \pm 0.002$ | $0.408 \pm 0.008$ | $0.470 \pm 0.006$ |
| $\alpha\beta$-C-GCN | $0.630 \pm 0.001$*** | $0.405 \pm 0.007$ | $0.476 \pm 0.006$*** |
| R-C-GCN | $0.657 \pm 0.001$*** | $0.424 \pm 0.007$** | $0.490 \pm 0.006$*** |
| SAGE | $0.711 \pm 0.002$ | $0.343 \pm 0.009$ | $0.539 \pm 0.004$ |
| $\alpha\beta$-C-SAGE | $0.721 \pm 0.002$*** | $0.352 \pm 0.009$** | $0.548 \pm 0.004$*** |
| R-C-SAGE | $0.734 \pm 0.002$*** | $0.360 \pm 0.008$*** | $0.551 \pm 0.004$*** |

## 5.3 REGRESSION WITH COUNT OUTCOME VARIABLES

We use two groups of datasets with count outcome variables. The first group consists of two Wikipedia datasets: Wiki-Chameleon and Wiki-Squirrel (Rozemberczki et al., 2019); both are page-page networks of Wikipedia pages with the visiting traffic as node labels. The second group is a co-citation network of papers at the EMNLP conferences. The goal is to predict the overall number of citations of each paper (including citations from outside EMNLP). We use the $R^2$-deviance, an $R^2$ measure for count data (Cameron & Windmeijer, 1996), to measure the model performance.

**Results.** The results of the count regression tasks are shown in Table 3. Intuitively, hyper-linked web pages or co-cited papers are more likely to be visited or cited together, therefore leading to correlated outcome variables captured by the graph. Indeed, we observe that the different variants of CopulaGNN outperform their base model counterparts in almost all setups. However, as the correlation may not be as strong as in the U.S. Election dataset, we observe that the regression-based parameterization (R-C-GCN and R-C-SAGE) has a greater advantage.

## 6 CONCLUSION

In this work, we explicitly distinguish the representational and correlational roles of the graph representation of data. We demonstrate through a simulation study that many popular GNN models are incapable of fully utilizing the correlational graph information. Furthermore, we propose CopulaGNN, a principled method that improves upon a wide range of GNNs to achieve better prediction performance when the graph plays both representational and correlational roles. Compared with the corresponding base GNN models, multiple variants of CopulaGNN yield consistently superior results on both synthetic and real-world datasets for continuous and count regression tasks.

### ACKNOWLEDGEMENT

Jiaqi Ma and Qiaozhu Mei were in part supported by the National Science Foundation under grant numbers 1633370 and 1620319.

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

## A  EXPERIMENT DETAILS

### A.1  DETAILS OF SYNTHETIC DATA GENERATION

We generate the synthetic data with the following procedure.

1. Sample a node feature matrix $\mathbf{X} \sim \mathcal{N}(\mathbf{0}, \boldsymbol{I}_{d_0})$, where $\mathbf{X} \in \mathbb{R}^{n \times d_0}$ and $d_0$ is the feature dimension.
2. Generate the graph in a way similarly to a latent space model (Hoff et al., 2002). First compute the latent variables $\boldsymbol{Z} = \mathbf{X}\boldsymbol{W}_g$, where $\boldsymbol{W}_g \in \mathbb{R}^{d_0 \times d_1}$ is a given weight matrix. Then calculate the latent distance $\|\boldsymbol{z}_i - \boldsymbol{z}_j\|_2$ for each node pair $(i, j), 1 \le i < j \le n$. Finally assign edges between the $m$ pairs of nodes with the shortest latent distances to form a graph with $n$ nodes and $s$ edges.
3. Assume $\boldsymbol{A}$ is the adjacency matrix of the graph, $\boldsymbol{D}$ is the degree matrix, and $\boldsymbol{L} = \boldsymbol{D} - \boldsymbol{A}$ is the graph Laplacian. Let $\tilde{\boldsymbol{A}} = \boldsymbol{A} + \boldsymbol{I}$ and $\tilde{\boldsymbol{D}} = \boldsymbol{D} + \boldsymbol{I}$. Given parameters $\boldsymbol{w}_y \in \mathbb{R}^{d_0}$, generate the node label vector $\boldsymbol{y} \sim \mathcal{N}(\boldsymbol{\mu}, \boldsymbol{\Sigma})$, where, for some $\gamma > 0, \tau > 0$, and $\sigma^2 > 0$,
   (a) $\boldsymbol{\mu} = \tilde{\boldsymbol{D}}^{-1}\tilde{\boldsymbol{A}}\mathbf{X}\boldsymbol{w}_y, \boldsymbol{\Sigma} = \sigma^2 \boldsymbol{I}$;
   (b) $\boldsymbol{\mu} = \mathbf{X}\boldsymbol{w}_y, \boldsymbol{\Sigma} = \tau(\boldsymbol{L} + \gamma\boldsymbol{I})^{-1}$;
   (c) $\boldsymbol{\mu} = \tilde{\boldsymbol{D}}^{-1}\tilde{\boldsymbol{A}}\mathbf{X}\boldsymbol{w}_y, \boldsymbol{\Sigma} = \tau(\boldsymbol{L} + \gamma\boldsymbol{I})^{-1}$.

### A.2  SIMULATION DETAILS FOR SECTION 3

For each configuration we randomly generate 100 datasets with different seeds. For each dataset, we randomly split the nodes into training, validation, and test sets equally to form a semi-supervised learning task.

We set the number of layers as 2 and the total hidden units as 16 for all models. We use the Adam optimizer (Kingma & Ba, 2015) with an initial learning rate of 0.01 to train all models by minimizing the MSE loss, with early stopping on the validation set. Finally, we report the $R^2$ score on the test set.

### A.3  EXPERIMENT DETAILS FOR SECTION 5

**Training details.** For all neural networks involved in the experiments, we set the number of layers as 2 and the number of hidden units as 16. We use the Adam optimizer to train all the models and apply early stopping on a validation set. For the real-world datasets, the initial learning rate is chosen from $\{0.01, 0.001\}$ on the validation set.

**The U.S. Election dataset.** The nodes in the election data are U.S. counties and edges connect adjacent counties on the map. Each county is associated with demographic and election statistics. In each of the regression tasks, one statistic is selected as the node outcome and the remaining statistics are used as the node features. The four regression tasks are named by the outcome statistics: Education, Election, Income, Unemployment. We randomly split the data into training, validation, and test sets with ratio 6:2:2 following Jia & Benson (2020), and we refer to their work for more details of the datasets.

**The Wikipedia datasets.** Each of the two Wikipedia datasets consists of a graph on Wikipedia, where each node is a Wikipedia page related to the animal of the page title and edges reflect mutual hyper-links between the pages. The node features are principal components of binary indicators for the presence of certain nouns. The count outcome variable of each node label is the monthly traffic in the unit of thousands.

**The EMNLP dataset.** This dataset is constructed from the DBLP citation data provided by AMiner (Tang et al., 2008). We first extract a set of papers published on the EMNLP conference and treat each paper as a node. Then we construct a graph where two papers have an edge if they are *cited simultaneously* by at least two EMNLP papers. The node features are principal components of the bag-of-words of paper titles and abstracts as well as the year of publication. The node label is the number of citations of each paper *from outside* EMNLP. For both types of datasets, we randomly split the data into training, validation, and test sets with ratio 1:1:1.

## B MORE DETAILS ABOUT COPULAS

### B.1 TWO-DIMENSIONAL EXAMPLES

Figure 2 shows PDFs of two-dimensional distributions constructed using different parametric copulas; the marginal distributions are all standard normal.

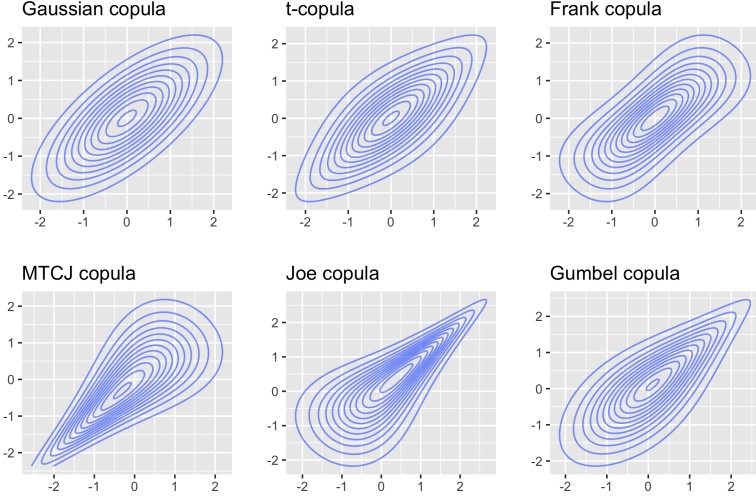

Figure 2: Density functions of two-dimensional distributions constructed using copulas. The marginal distributions are all standard normal. See Joe (2014) for the definitions of these parametric copulas.

### B.2 APPROXIMATING THE COPULAS FOR DISCRETE RANDOM VARIABLES

If the random vector $\boldsymbol{Y}$ is discrete, the copula representation of its PMF is more complex. Take $n = 2$ as an example, the PMF of $\boldsymbol{Y} = (Y_1, Y_2)$ is

$$
\begin{aligned}
f(\boldsymbol{y}) &= \mathbb{P}(Y_1 = y_1, Y_2 = y_2) \\
&= \mathbb{P}(Y_1 \le y_1, Y_2 \le y_2) - \mathbb{P}(Y_1 < y_1, Y_2 \le y_2) - \mathbb{P}(Y_1 \le y_1, Y_2 < y_2) + \mathbb{P}(Y_1 < y_1, Y_2 < y_2) \\
&= C(u_{12}, u_{22}) - C(u_{11}, u_{22}) - C(u_{12}, u_{21}) + C(u_{11}, u_{21}),
\end{aligned}
$$

where $u_{i1} = \lim_{x \to y_i^-} F_i(x) = F_i(y_i^-)$ and $u_{i2} = F_i(y_i)$. In general, the PMF has the following form:

$$
f(\boldsymbol{y}) = \sum_{j_1=1}^{2} \cdots \sum_{j_n=1}^{2} (-1)^{j_1 + \cdots + j_n} C(u_{1j_1}, \ldots, u_{nj_n}), \tag{5}
$$

which is computationally intractable because there are $2^n$ summands. Kazianka & Pilz (2010) propose an approximation of the above PMF based on the generalized quantile transform. It smooths the CDF of ordinal discrete variables from a step function to a piece-wise linear one:

$$
f(\boldsymbol{y}) \approx c(v_1, \ldots, v_n) \prod_{i=1}^{n} f_i(y_i), \tag{6}
$$

where $v_i = (u_{i1} + u_{i2})/2$. It has been shown that the approximation works well as long as the marginal variance is not too small. We apply this method to approximate PMF to handle discrete random variables.

