# OpenReview forum: "CopulaGNN: Towards Integrating Representational and Correlational Roles of Graphs in Graph Neural Networks"
_ICLR.cc/2021/Conference — ICLR 2021 Poster_

### Official Review · AnonReviewer2 · 2020-10-27
**An interesting work, but the way of decomposing joint distribution may not be elegant**

**Rating:** 5
**Confidence:** 3

**Review:**

This paper proposes to distinguish the representational and correlational roles of the graph representation of data. The authors proposes CopulaGNN that decomposes the joint distribution of node labels into the product of marginal densities and a copula density, which models the representational information and correlational information separately. The experimental results demonstrate the effectiveness of the proposed method. The idea of introducing copula into GNN is novel and interesting, and this paper is basically well written and well organized. However, I have the following concerns:

1. Characterizing representational role and correlational role is the key motivation of the proposed method, but I'm not quite clear about their definitions. As far as I understand, the representational role means graph homophily and the correlational role means graph topology. But homophily and topology are not two disjoint concepts since homophily says that two connected nodes are likely to have the same feature, which is exactly based on (local) topology. I think a better way to decompose the model is to say that, the label of a node depends on not only its feature but also the graph topology.

2. The authors decompose the joint distribution of node labels into marginal distributions of node labels as well as a copula density. The marginal distributions characterize the representational information (homophily) and the copula density characterizes the correlational information (topology). Here the marginal distribution f_i(y_i; X, G) is not only dependent on its own feature but only the feature of its neighbors, which is not reasonable since the marginal distribution should only characterize the node itself. This is also connected to the above argument that the dependence of node label can be decomposed into its feature and graph topology. Therefore, it seems more reasonable to write eq. 2 as f(y; X, G) = c(u_1, ..., u_n; G) \prod f_i (y_i; X). Only in this way we can say that we "decompose" the joint distribution of node labels. The original eq.2 does not decompose anything since X and G appear in both copula density and marginal distributions. I'm not saying that the current eq.2 do not work, but it "seems" more elegant to do as what I said above. Otherwise the original eq.2 is just like fitting the joint distribution with two complete modules then simply combining them together.

3. Another issue that confuses me is that, the studied problem is formulated as learning joint distribution of node labels, but we actually only have one observed sample, i.e., the whole graph. Although the authors have tried their best to reduce the parameters of the model, it is still prone to overfitting if you maximize the likelihood only on one sample.

4. Modeling the problem as learning joint distributions will cause another issue that doing inference is too time confusing, since we need to infer labels of all nodes as the same time even if we only need the label of one node. In addition, as the authors said, multiple sampling rounds may be required to calculate the average.

---

> ### Author Response · Authors · 2020-11-13
> **Response to AnonReviewer2**
>
> We thank the reviewer for the detailed comments and we list our responses below.
>
> ---
>
> Response to 1 and 2.
>
> We first note that f(y; X, G) is the *conditional* joint distribution of node labels conditioned on the graph G and node features X. So for any decomposition of the conditional joint distribution, everything should still be conditioned on G and X unless specific dependence assumptions are made. By Sklar's theorem, we can write ANY conditional joint distribution of the node labels in the form of Eq. 2. But this will no longer be true if we replace the marginal distributions and the copula with f_i(y_i;X) and c(u_1,...,u_n;G).
>
> Second, we highlight that, while both the marginal distributions f_i(y_i;X,G) and the copula c(u_1,...,u_n;X,G) are functions of X and G, they are not interchangeable "complete modules" and capture drastically different things. The product of marginals \prod f_i(y_i;X,G) does not capture the conditional dependencies among the node labels so the learning objectives will be simple; while the copula is designed to capture the conditional dependencies only so it can have a simple form.
>
> Finally, we clarify (with a paraphrase of the third paragraph of our Section 1) that graph topology information is used in both the representation role and the correlational role. The representational role refers to graph information that can be used to construct better node representations, which includes but is not limited to feature aggregation (homophily) and the (local) topology. The correlational role, on the other hand, captures the conditional dependencies of the node labels conditioned on the node representations. The conditional dependencies are assumed to be related to the graph topology in our model. However, the use of the graph topology in the correlational role is orthogonal to that used in the representational role.
>
> ---
>
> Response to 3.
>
> This is a good question. Overfitting may occur if we do not put constraints on the precision matrix. However, our proposed parametrization for the precision matrix reduces the number of parameters significantly to O(1) in terms of the number of nodes and hence effectively prevents overfitting issues.
>
> In addition, our learning problem is similar with that in the well-studied random effects model [1]. In a random effects model, there are n data points (x_i, y_i), i=1,...,n. The outcome variables y_i could be dependent due to the subpopulation structure in the data, and the outcome variables are associated with a covariance matrix with unknown parameters to be learned. So the learning setup is similar with ours: one sample of each y_i and covariance with parameters to be learned. Also similarly, the number of parameters remains O(1) in terms of n due to some dependence assumptions, and the overfitting is not an issue.
>
> ---
>
> Response to 4.
>
> We acknowledge that modeling the joint distribution indeed increases the inference time. However, if one only cares about one or few nodes, we can simply drop other missing-label nodes in the conditional distribution of Eq. 3, thanks to the easy marginalization of multivariate Gaussian [2].
>
> Further, for large graphs, there are often block/community structures. A simple trick to reduce the computation cost is to enforce block structures in the precision matrix so that the inference of a node will only depend on the nodes in the same block. With this trick, we can also have a smooth trade-off between the computation cost and the capability of capturing the correlational information.
>
> ---
>
> References
>
> [1] Wooldridge, J.M., 2010. Econometric analysis of cross section and panel data. MIT press. pp 257--264
>
> [2] https://en.wikipedia.org/wiki/Multivariate_normal_distribution#Marginal_distributions

---

### Official Review · AnonReviewer1 · 2020-10-28
**A useful perspective on correlation on GNNs**

**Rating:** 7
**Confidence:** 4

**Review:**

Summary:

This paper proposes a novel framework for the use of graph neural networks for node-level prediction when these predictions have correlations across nodes that cannot be entirely explained by the node features. In a nutshell, the idea is to propose a probabilistic model for the joint node-level output whose mean can be parametrized as a GNN and whose correlation is determined by a precision matrix with the same support as the graph at hand.

Overall:

My evaluation is positive. Although the presentation could be streamlined and made clearer (see comments below), the explicit recognition for the need of a learnable module for the correlation that is separated from the GNN seems to be a fundamental insight that can shed light on multiple existing methods.

Pros:

1 - The synthetic example in Section 3.2 nicely illustrates the shortcoming of some existing methods.

2 - The generic framework is applicable on top of existing GNNs architectures. As such, it can be applicable to a wide range of settings

Cons:

1 - The presentation is hard to follow at some points. For example, the “Model Inference” paragraph is key for the implementation of the proposed method. However, it is presented in only a handful of lines without any discussion on the different steps. Maybe even an Algorithm environment would be preferred for this.

2 - It is unclear to me the true value in doing the presentation around copulas. Section 4.1 is too short in the sense that it would be difficult for someone that does not know what a copula is beforehand to actually pick up the necessary notions from there. The key idea, formally stated in Section 4.2, can be explained without resorting to copulas, simply in terms of marginal distributions and covariances. What is exactly the value of introducing the copula formalism? Please make more emphasis on this.

++++++++++++++++++++++++++++++++++++++++++++

Edit after author response: We thank the authors for their response to my concerns and those of other reviewers.

---

> ### Author Response · Authors · 2020-11-13
> **Response to AnonReviewer1**
>
> We appreciate the constructive suggestions by the reviewer and we updated the draft accordingly.
>
> In particular, we added an algorithm table for the details of the model inference and expanded Section 4.1 a little bit. The necessity of Section 4.1 is to support Eq. 2 with Sklar's theorem, i.e., the joint distribution can be decomposed into the marginals and the copula. Section 4.1 also defines some important notations such as u_1, ..., u_n.

---

### Official Review · AnonReviewer4 · 2020-10-30
**Deep insights into the Graph Neural Networks**

**Rating:** 7
**Confidence:** 4

**Review:**

This paper proposes to use Copula function to model the correlational role in the node-level prediction tasks. From my own perspective, I quite enjoy this work. In particular, it uses a simple simulation example to uncover the importance of correlational role, i.e., most of the GNNs approaches might perform worse than simple MLPs as they are incapable of using the correlation graph information. This observation is simple, however, would provide deep insights into the current GNNs literature.

[Introduction & Related Work]
The authors clearly state their motivation and precisely introduce the mentioned works. Readers would get a clear understanding on the background information.

[Simulation the two roles]
The authors use simple simulation examples to show the importance of correlation graph information.

[Copula Graph Neural Networks]
The authors introduce the detail architecture design for the model. Although the design looks reasonable, I have several questions as follows:

1, the authors mainly used the Gaussian Copula as the correlation modeling choice and incorporate the graph structure in the precision matrix. I can totally understand that this setting can reduce the number of parameters. What if the precision matrix is without the graph structure and is simply placed with sparse priors?

2, following the above question, it looks like Gaussian Copula is the only suitable choice for Copula functions. How can one use other copula functions, as we cannot incorporate the graph structure and the other copula functions needs complex structures (e.g. C-vines or D-vines) to model the high dimensional data?

3, the two-parameter and regression parameterisation seems quite complicate in fitting the precision matrix. Thus, can these parameterisation method also be used for other copula functions?

[Experiments]
The experimental part can be sufficient to verify the effectiveness of the authors' proposed model.

---

> ### Author Response · Authors · 2020-11-13
> **Response to AnonReviewer4**
>
> We appreciate the great questions raised by the reviewer and we address them in detail below.
>
> ---
>
> Response to 1.
>
> We first note that we investigate the correlational role of the graph in this paper and it is our assumption in the proposed CopulaGNN that the sparsity of the precision matrix is related to the graph. In practice, there are certainly cases where other sparsity assumptions are more suitable for the data. In these cases, we need to change the parameterization of the copula to accommodate those assumptions.
>
> ---
>
> Response to 2.
>
> Thanks for the great question. A generalization of the Gaussian copula family is the elliptical copulas, i.e., copulas of elliptical distributions. Like the Gaussian copula, an elliptical copula is parameterized by a correlation matrix, which makes them compatible with the proposed method. One common example of elliptical copulas is the t-copula, which is the copula of the multivariate t-distribution. Compared with the Gaussian copula, t-copulas are more suitable when tail dependence is present. See [1] and references therein for an introduction to elliptical copulas.
>
> As a future research direction, we plan to generalize the method beyond elliptical copulas. We agree with the reviewer that it is not easy to build a vine structure from a graphical model, but we might get inspiration from the literature that tries to link vines with graphical models, for example, [2].
>
> ---
>
> Response to 3.
>
> As mentioned above, those parametrizations could be generalized to other elliptical copulas. But it is yet unclear to us how to apply them to other copula families and we will devote more effort along that direction in the future.
>
> ---
>
> References
>
> [1] http://www.columbia.edu/~rf2283/Conference/2Models%20(1)%20Seagers.pdf, pp 45--51.
>
> [2] Haff, I. H., Aas, K., Frigessi, A., & Lacal, V. (2016). Structure learning in Bayesian Networks using regular vines. Computational Statistics & Data Analysis, 101, 186-208.

---

### Official Review · AnonReviewer3 · 2020-10-31
**Graphical Neural Network using copulas**

**Rating:** 7
**Confidence:** 3

**Review:**

##########################################################################

Summary:

The paper presents a new model based on the Graphical Neural Network (GNN). The proposed model adopts probability distributions called copulas and is called the Copula Graphical Neural Network (CopulaGNN). Two parametrizations of the CopulaGNN are given, and the learning of the proposed model is discussed. Experiments suggest that the CopulaGNN outperforms existing GNNs and MLP in almost all setups.


##########################################################################

Reasons for score:

Unlike many existing GNNs which focus only on the representational role, the proposed CopulaGNN can be used for the modelling of not only representational information but also correlational information. My concern is that, although various families of copulas exist, it is not clear why the authors consider only the Gaussian copula. This is not consistent considering the fact that multiple types of (marginal) outcome variables are discussed in the paper.

##########################################################################

Pros:

(1) Many existing GNNs focus only on the representational role. Some methods are available for modelling the dependence of variables (or correlational information). The proposed method is nice because, unlike most existing methods, the CopulaGNN enables us to model both the representational and correlational information. This is due to the advantage of copula modelling in which the joint distribution of node outcomes can be decomposed into the product of marginal densities and a copula density.

(2) Copula is a probability distribution which can be used for flexible modelling. Because of this benefit, the proposed CopulaGNN can also be used as a flexible model.

(3) The experiments consider both discrete and continuous outcome variables both of which potentially appear in practice.

##########################################################################

Cons:

(1) Many families of copulas have been proposed in the literature. However the paper considers only the Gaussian copula. This is not a consistent approach because the authors consider multiple types of (marginal) outcome variables in the paper.

(2) Considering that the GNN has a graph structure, one possible choice is to use the vine copula instead of the Gaussian copula as a copula for the GNN. Since the vine copula is a multivariate copula with graphical structure, this copula might match well with the GNN in terms of theoretical and interpretational aspects.

(3) It is known that the learning of the copula is not unique if outcome variables are discrete. Does this fact cause any problem in the experiments in Section 5.3?

##########################################################################

Questions during rebuttal period:

Please address and clarify the cons above.

#########################################################################

Typos:

(1) p.4, Section 3.3, l.1: Since $n$ is assumed to be greater than $m$ in Section 3.1, it does make sense to assume $n=300$ and $m=5000$ here.

(2) p.6, l.13 up: Equation 1 -> Equation (1)


---

### Updates:

Thanks for the authors' response. My major concerns were related to the comments (1) and (2) in Cons, but it is now clear why the authors consider only the Gaussian copula in the paper. My other concerns are addressed, too. I upgraded my rating after reading the authors' response.

---

> ### Author Response · Authors · 2020-11-13
> **Response to AnonReviewer3**
>
> We thank the reviewer for the insightful questions and we answer your specific questions below.
>
> ---
>
> #### (1) The relationship between the choice of copula and the outcome types.
>
> We would like to clarify that the purpose of using copulas is to decompose a joint distribution; in other words, conceptually we have the following decomposition:
>
> joint distribution = copula + marginal distributions
>
> Therefore, the choice of the copula family is *independent* from that of the marginal distributions. For example, in our work, we use (Gaussian copula + univariate normal) for continuous labels, and (Gaussian copula + Poisson) for discrete labels. Other marginal distribution families can also be combined with the Gaussian copula if needed.
>
> The reason we use the Gaussian copula is that its parametrization has a natural connection with graphical models. The Gaussian copula is parametrized by a covariance matrix, as defined in Section 4.1, and the inverse of the covariance matrix corresponds to a graphical model, as illustrated in Section 4.2.
>
> Other than the Gaussian copula, commonly used high-dimensional copula families include Archimedean copulas and vine copulas. However, Archimedean copulas are permutation symmetric, which dramatically limits their applicability. A short introduction to Archimedean copulas is given in [1]. Vine copulas also have their limitations, which we will discuss next.
>
> ---
>
> #### (2) How about vine copula?
>
> We appreciate that the reviewer brought up the connection between graphical models and vine structures. However, vines are more complex and restricted than a graphical model. A vine is a sequence of trees; the edges in the first tree are the nodes of the second tree, the edges of the second tree are the nodes of the third tree, etc. It also needs to satisfy the so-called proximity condition and simplifying assumption; see [2] for a detailed exposition. Therefore, as mentioned by AnonReviewer4 as well, there is no straightforward correspondence between a graphical model and a vine structure.
>
> As a future research direction, we would like to explore the possibility of generalizing our method from the Gaussian copula to vine copulas. A good starting point might be the literature that tries to link vines with graphical models, for example, [3].
>
> ---
>
> #### (3) How to deal with discrete outcomes?
>
> We note that we described the treatment for the discrete outcomes in Appendix B.2 in our original submission. When the outcome is discrete, we use an approximation proposed by Kazianka & Pilz (2010) for the copula and it works well in practice.
>
> ---
>
> #### (4) Typos
>
> 1) Thank you very much for the detailed suggestion! We apologize that we overloaded the notation $m$. In section 3.1, $m$ represents the number of labeled nodes while in section 3.3, $m$ represents the number of edges. We changed the number of edges as $s$ in the updated draft.
>
> 2) We corrected it in the updated draft. Again, thanks for catching it!
>
> ---
>
> #### References
>
> [1] http://www.columbia.edu/~rf2283/Conference/2Models%20(1)%20Seagers.pdf, pp 3--23.
>
> [2] Kraus, D., & Czado, C. (2017). Growing simplified vine copula trees: improving Di {\ss} mann's algorithm. arXiv preprint arXiv:1703.05203.
>
> [3] Haff, I. H., Aas, K., Frigessi, A., & Lacal, V. (2016). Structure learning in Bayesian Networks using regular vines. Computational Statistics & Data Analysis, 101, 186-208.

---

### Comment · ~Wei_Qiu3 · 2021-02-09
**Release of the Source Code**

Dear Authors,

I am very interested in this idea by using copula. Are you planning to release the source code?

Thank you for your attention.

Best regards,

Wei.

---

> ### Author Response · Authors · 2021-02-09
> **Thanks for your interest!**
>
> Yes, we are releasing the code soon.

---

> > ### Comment · ~Wei_Qiu3 · 2021-05-31
> > **Possible Typos**
> >
> > Thanks, I have cloned your code. After reading the paper one more time. I found there maybe 2 typos in your paper.
> >
> > 1. In Sec. 1 the first contribution
> >
> > > We raise the question of distinguishing the two roles played by the and
> >
> > It seems that the object following the "the" is missing. I think the object is "graph".
> >
> > 2. In Algorithm 1, line 13,
> >
> > Should the $z_i$ be $z^l_i$? If not, what is the purpose of $z^l_i$?
> >
> > Thanks in advance.

---

> > > ### Comment · ~Jiaqi_Ma1 · 2021-05-31
> > > **Thanks for spotting them!**
> > >
> > > Hi Wei,
> > >
> > > Yes, you are right. Those are typos. I have updated the paper. Thanks for spotting them!

---

> > > > ### Comment · ~Wei_Qiu3 · 2021-09-28
> > > > **You are Welcome.**
> > > >
> > > > Dear Jiaqi,
> > > >
> > > > You are welcome. CopulaGNN is a good idea, it inspired me a lot.
> > > >
> > > > Best regards,
> > > > Wei.

---

### Decision · Program_Chairs · 2021-01-07
**Final Decision**

**Decision:**

Accept (Poster)

**Comment:**

Three referees support accept and one indicates reject. The issues pointed out by the reviewer who proposed rejection should be properly reflected in the final version.

First, regarding the synthetic experiment that illustrates the shortcomings of the existing GNN models, three reviewers, including myself, judged quite interesting. However, note the opinion of one reviewer that it is more appropriate to separate the influence of feature x and graph structure in the label generation method and each independently contribute to label generation. This part should be more justified in the final version.

In addition, it was pointed out that the expressive power of the model may be limited according to the parameterization type of the precision matrix, and there is a limitation that there may be a disadvantage in inference because it is a copula-based probabilistic model. I think this characteristic is actually a fundamental limitation of the proposed method. However, three reviewers, including myself, thought that it was an interesting framework as a role that can complement the message passing architecture, and decided that the possibility of the proposed method was worth publishing. However, in order to reinforce this argument a little more, it would be better if the final version verifies it with more diverse GNN architectures and datasets.

---

> ### Author Response · Authors · 2021-03-17
> **Thank you for your support for the acceptance!**
>
> We appreciate your support for the acceptance of this paper and we address the concerns raised in the meta-review below.
>
> ---
>
> (1) The influence of feature and graph.
>
> AnonReviewer2 seems to mis-interpret the concepts of the two roles we defined in the paper. We would like to clarify that these two concepts do not separate the influence of feature and graph. Rather, both the representational role and the correlational role are about the influence of the graph, but in two distinct ways. We provided some more detailed explanations in our earlier response to AnonReviewer2. To make it clearer, we have also revised the relevant paragraph to highlight this point in the camera-ready version.
>
> ---
>
> (2) Expressive power of the parameterization.
>
> First, we would like to highlight that the regression-based parameterization of the proposed method utilizes an MLP to parameterize each non-zero entry of the precision matrix, which is fairly expressive. The number of parameters is on par with the base GNN in this case. In addition, we note that the proposed method is an orthogonal augmentation to most existing GNNs so we do not lose the expressive power in the base GNNs.
>
> Second, we want to point out that there is a trade-off between expressive power and estimation accuracy, which is limited by the number of observations of node labels. In the experiments, we indeed observe that occasionally the simpler two-parameter parameterization performs better than the more expressive regression-based parameterization.
>
> ---
>
> (3) Computation cost of inference.
>
> Besides our earlier response to AnonReviewer2, we would like to add additional remarks that, thanks to the sparse precision matrix, the computation cost can be significantly reduced to about linear in the graph size, with numerical tricks. We have added a discussion on this topic in the camera-ready version.